# Recent community warming of moths in Finland is driven by extinction in the north and colonisation in the south

Emilie E. Ellis [1] ✉, Laura H. Antão [1,2], Andréa Davrinche [1], Jussi Mäkinen[1,3], Mark Rees [4], Irene Conenna[1], Ida-Maria Huikkonen[3], Reima Leinonen [5], Juha Pöyry [3], Anna Suuronen[3], Anna-Liisa Laine [1], Marjo Saastamoinen [1], Jarno Vanhatalo [1,6] & Tomas Roslin [1,7]

As the climate warms, species are shifting their ranges to match their climatic niches, leading to the warming of ecological communities (thermophilisation). We currently have little understanding of the population-level processes driving this community-level warming, particularly at rapidly warming high latitudes. Using 30 years of high-resolution moth monitoring data across a 1200 km latitudinal gradient in Finland, we find that higher latitude communities are experiencing more rapid thermophilisation. We attribute this spatial variation to colonisation-extinction dynamics, both for the full community and for thermal affinity groups. Our findings reveal that latitudinal variation in the pathways underpinning thermophilisation is the net outcome of opposite forces: in the north, community warming is driven by the extinction of cold-affiliated species, while in the south it is driven by high colonisation rates of warm-affiliated species. Thus, we show how species' thermal affinities influence community reorganisation and highlight the elevated extinction risk among cold-affiliated species.

Global warming is causing widespread climate-induced shifts in biotic and abiotic conditions[1]. Such changes are simultaneously rendering habitats within a species' current range intolerable and expanding the availability of suitable conditions to new areas. Species display a diversity of non-random responses to these changes, with variation among species driven by their traits[2,3]. In this context, metrics of species' thermal niches have proven a powerful predictor of responses to global warming[4–6]. At the community level, there is strong evidence of a shift towards warm-affiliated species, which is reflected in a higher community mean of species-level thermal niches, referred to as thermophilisation[7].

A species' thermal niche is often summarised as the average temperature within its range (the Species Temperature Index, STI)[8] and the community-level response as the abundance-weighted mean of the STIs (Community Temperature Index, CTI[9]). Broadly, thermophilisation can occur through three pathways based on species colonisation and extinction patterns. These pathways are also linked to changes in the number of individuals and can be further influenced by the available species pool (Fig. 1)[5]. Community warming may stem from local extinctions of cold-affiliated species, as suitable climatic conditions become unavailable, leading to reductions in species richness[10]. Conversely, increases in species richness may result from the colonisation of warm-affiliated species, which have shifted their distributions to align with their optimal temperature range[11]. Lastly, species richness may remain stable even as community composition changes due to high

[1]Research Centre for Ecological Change, Organismal and Evolutionary Research Programme, Faculty of Biological and Environmental Sciences, University of Helsinki, Helsinki, Finland. [2]Department of Biology, Faculty of Science, University of Turku, Turku, Finland. [3]Finnish Environment Institute (SYKE), Helsinki, Finland. [4]School of Biosciences, University of Sheffield, Sheffield, UK. [5]Kainuu Centre for Economic Development, Transport and the Environment, Kajaani, Finland. [6]Department of Mathematics and Statistics, Faculty of Science, University of Helsinki, Helsinki, Finland. [7]Department of Ecology, Swedish University of Agricultural Sciences (SLU), Uppsala, Sweden. ✉e-mail: emilie.ellis@helsinki.fi; emilie.ellis95@gmail.com

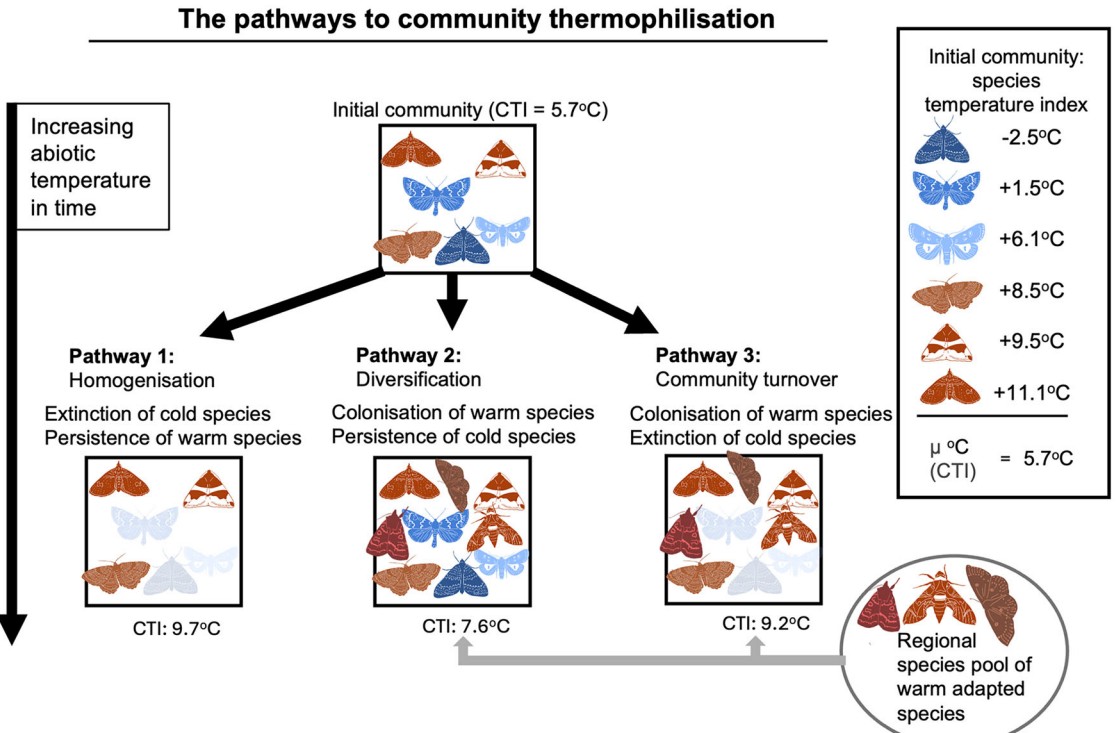

**Fig. 1 | Conceptual Figure showing the simplified pathways in which communities may experience community warming in response to increasing abiotic temperature through time (i.e. thermophilisation as measured by increases in CTI).** In pathway 1, species richness decreases as cold-affiliated species selectively go locally extinct. In pathway 2, species richness increases through colonisation by warm-affiliated species. In pathway 3, species richness remains stable, while the identity of species changes via the substitution of cold-affiliated species by warm-affiliated species. Species' thermal affinities (measured as a species' temperature index, STI) are indicated by their colouring (red for warm, blue for cold) and species local extinctions are shown by semi-transparency.

species turnover[7,12,13]. In this case, communities undergo transformation through the displacement of cold-affiliated species by warm-affiliated ones. Hence, a community's response to warming can be attributed to the ability of species to track their climatic niches through colonisation, as well as their ability to persist in a particular environment.

Given the widespread effects of temperature change on species and communities[5,9,14] and the increasing availability of large-scale distributional and abundance time series data (e.g. Global Biodiversity Information Facility (GBIF, https://www.gbif.org/) and BioTIME;[12]), there is a growing effort to estimate STIs (e.g. Moore et al. [15]) to assess patterns of thermophilisation across terrestrial, marine and freshwater communities[5,9,16]. However, given the overall focus on the emergent patterns at the community level, a thorough understanding of the underlying mechanisms of thermophilisation is lacking[5,17]. Importantly, the drivers of community warming are likely to vary in space. Such context-dependence may emerge from the following: first, climate warming is occurring at different rates in different regions[18]. Second, local species richness is constrained by the regional pool of species[19]. Due to dispersal and niche limitation, the species reaching a local community will mainly consist of species present in the wider region. Thus, the availability of species with higher thermal affinities to replace species with cooler thermal affinities will vary in space. Here, latitude offers a key gradient to capture these impacts (Fig. 1). Along the latitudinal gradient towards the Arctic, mean annual temperatures decline[20], climate warming intensifies[21], species richness declines[22] and thermal trait distributions shift towards colder temperatures[23]. How this combination of environmental and ecological factors shapes the pace at which communities are changing is largely unexplored. Furthermore, previous attempts to examine population-driven changes in community warming have largely focused on marine[16], bird[24] and plant[25] communities.

Here, we present a comprehensive assessment of the processes driving changes in community thermal composition and the spatial variation in these patterns. Our analysis is based on high-resolution spatiotemporal data collected over 30 years through a systematic monitoring programme of moths (Insecta: Lepidoptera) across 62 sites in Finland, spanning latitudes from 59° to 70° North. We calculate STI and CTI using range maps (cf.[8,26]), and implement an incidence-based approach to estimate colonisation and extinction rates[27,28] to address three research questions:

1. Are northern moth communities experiencing accelerated thermophilisation?
2. How do colonisation-extinction dynamics vary with latitude?
3. How does the thermal affinity of species influence community thermophilisation patterns?

Since climate change and its ecological consequences tend to be more severe at higher latitudes[14,29,30], we hypothesise that community warming will be accentuated towards the north (Supplementary Fig. 1). As Finland represents the northern edge of the European landmass, we also predict that local extinction rates will be higher in the north—and particularly high for cold-affiliated species, as they cannot track their climate further north[7,10,31]. Conversely, we predict that the inflow of species from large species pools in mainland Europe will result in higher colonisation rates of warm-affiliated species in the south of Finland. As the net outcome of latitudinal variation in species turnover, we expect a declining gradient in the relative contribution of colonisation by warm-affiliated species vs. extinction of cold-affiliated species to community thermophilisation from southern to northern Finland.

## Results
### Are northern communities experiencing faster thermophilisation?
Linear mixed models revealed an increase in the Community Temperature Index (CTI) of moth communities through time, with a

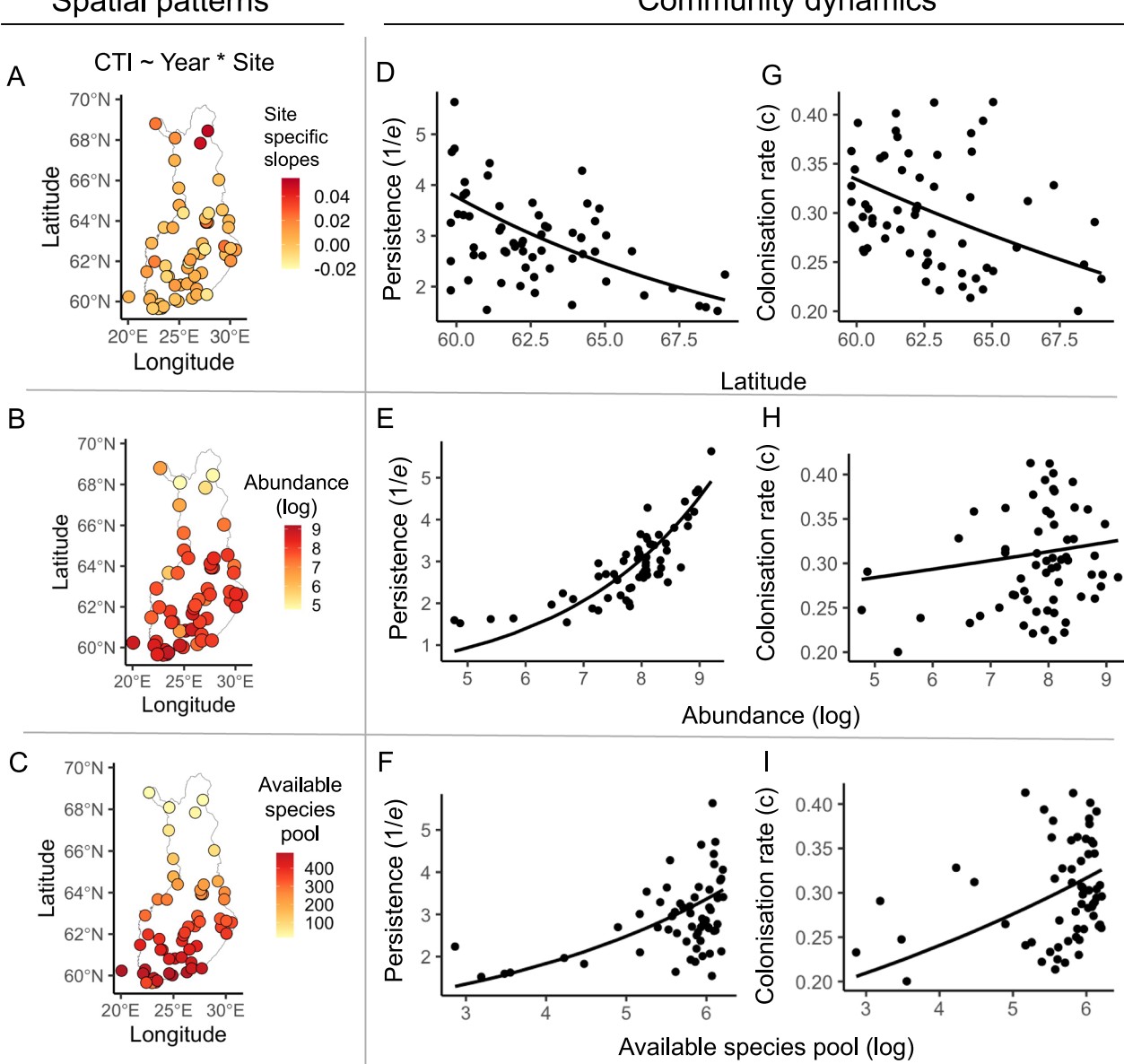

**Fig. 2 | Patterns in the biotic and abiotic factors influencing colonisation rates and persistence of moth communities.** The left-hand panels show spatial patterns across Finland in **A** changes in CTI through time, **B** the abundance (i.e. total number of individuals observed at a site averaged across years) on a logarithmic scale, and **C** the available species pool (i.e. the total number of species in the pool averaged across years). The middle panels show the relationships between persistence and **D** latitude, **E** abundance **F** available species pool, while the right-hand panels show the relationships between colonisation rates and **G** latitude, **H** abundance, and **I** available species pools. Each point is a site-specific observation ($n = 62$). The lines in panels **D–I** show the model fits using a maximum-likelihood modelling approach (see Supplementary Table 2 for full model outputs).

steeper slope in the north ($\chi^2 = 38.23$, df = 2, $p < 0.001$, Fig. 2A, Supplementary Table 1, Fig. 2).

**How do colonisation-extinction dynamics vary with latitude?**
We used stochastic demographic colonisation-extinction models to quantify community dynamics[27]. These models are rooted in the Theory of Island Biogeography, which states that the species richness of an island's community is determined by the rate at which species colonise and go extinct[32]. For each site, we estimated a colonisation rate ($c$: $1/c$ is the average time (years) to colonisation) and an extinction rate ($e$: $1/e$ is the average time to extinction). Overall, our colonisation-extinction models showed that colonisation rates were lower and less variable than extinction rates ($c$ range = 0.20–0.40, $e$ range = 0.17–0.65). Across our dataset, the average time it took for a

moth species to colonise a site was 3.3 years compared to 2.7 years for an extinction within a site, and these rates varied with both abiotic and biotic factors. Latitude, abundance (i.e. the total number of individuals per site averaged across years) and the size of the available species pool (i.e. the total number of species that could colonise a site based on their distribution averaged across years) at each site were important predictors of persistence ($1/e$) and colonisation rates (Fig. 2 and Supplementary Table 2). Moth abundance and available species pools both decreased with increasing latitude (Fig. 2B, C). At the same time, latitude was strongly negatively correlated with both persistence (Fig. 2D, slope estimate = −0.85, $p < 0.0001$) and colonisation rates (Fig. 2G, slope estimate = −0.37, $p < 0.0001$). Persistence increased with local moth abundance (Fig. 2E, slope estimate = 0.32, $p < 0.00001$). However, the effect of abundance on colonisation rates

was much shallower and slightly positive (Fig. 2H, slope estimate = 0.03, $p < 0.0001$). With a larger available species pool, both persistence (Fig. 2F, slope estimate = 0.30, $p < 0.0001$) and colonisation rates (Fig. 2I, slope estimate = 0.14, $p < 0.0001$) increased.

We then conducted the same modelling as described above, but partitioning the Finland-wide moth community into four quartiles based on the species' STIs. This resulted in species being split into four groups according to their thermal affinities: cold-adapted, cold-tolerant, warm-tolerant, and warm-adapted (Supplementary Fig. 4). Using these groupings, we generated colonisation-extinction rate pairs for each thermal group at each site[27]. The four thermal affinity groups exhibited distinct latitudinal ranges (Fig. 3 and Supplementary Table 3), and at their northern latitudinal limits, each thermal group showed reduced abundance and smaller available species pools (Fig. 3A, B), with correspondingly lower persistence and colonisation rates at their latitudinal edges (Fig. 3C, F).

Each thermal group's colonisation rates and persistence responded differently to the latitudinal gradient ($c$ slope range = −0.05 to −1.1 and $1/e$ slope range = −0.63 to −2.52, Fig. 3C, F). Nonetheless, the relationship between abundance and persistence was similar across thermal groups, as indicated by the smaller range in the slope estimates across groups (slope range = 0.24 to 0.35, Fig. 3D). This consistency suggests that community persistence is primarily driven by abundance, regardless of thermal affinity. There was only a weak relationship between abundance and colonisation rates, indicated by shallow slopes (slope range = −0.05 to 0.02, Fig. 3G). Additionally, both colonisation rates and persistence increased with the available species pool, with similar slopes across groups ($c$ slope range = 0.2 to 0.3, $1/e$ slope range = 0.3 to 0.4, Supplementary Table 3). This implies that larger species pools enhance both persistence and colonisation, irrespective of thermal affinity.

### What are the pathways to thermophilisation?

To resolve the different pathways leading to community warming (Fig. 1), we tested how variation in colonisation-extinction dynamics across thermal affinity groups and space contributes to community thermophilisation. To do this, we explored several components with a potential impact on extinction and colonisation dynamics, namely evaluating extinction rates, colonisation rates, species richness (as a descriptor of species observed at the site), species saturation (as a metric of potential colonists), and resulting net rates of colonisation and extinctions (i.e., the average number of species leaving or joining a community per year) for each thermal group and across three bioclimatic zones (South-, Middle- and North Boreal). Extinction rates ($e$), as previously observed (Figs. 2 and 3), were up to three times higher in the northern boreal sites (Fig. 4A). We found lower species richness ($S$) in the north and at the latitudinal edges of the thermal groups within bioclimatic zones (Fig. 4B, concurrent with patterns previously observed in the available species pools; Fig. 3A). Next, we used $S$ to estimate net extinction rates ($e(S)$: average number of species lost per year). We found that net extinctions were 2-fold higher in the south compared to the north (Fig. 4C). However, it should be noted that the fraction of species lost per year is higher in the north (Fig. 4A). Colonisation rates declined from cold- to warm- thermal affinity groups within bioclimatic zones, with warmer affiliated species having lower colonisation rates (Fig. 4D). Potential colonists ($Sp$-$S$: available species pool minus mean species richness) varied widely in space. In the south, there were a higher number of potential colonists compared to the north. We found that there were no potential colonists available from either the warm-adapted or warm-tolerant thermal affinity groups in the north boreal bioclimatic zone (Fig. 4E). Patterns of net colonisations ($c(Sp$-$S)$) were similar to the net extinctions. Nonetheless, net colonisation rates were generally lower than the net extinctions, especially in the north (Fig. 4C, F).

To examine how the observed variation in net colonisations and net extinctions contributed to thermophilisation (Fig. 1), we estimated the species richness at equilibrium, which is the balance between $e(S)$ and $c(Sp$-$S)$[27,33]. We multiplied the predicted species richness at equilibrium by the mean STI of each group (see 'Methods': Model Fitting and Inference) to predict the CTI values at each site ($CTI_{pred}$). When $CTI_{pred}$ is regressed on the observed CTI ($CTI_{obs}$), communities at equilibrium will fall along a line with a slope of 1. Communities above the line are cooler than expected, while communities below the line are warmer than expected. Here, we found that relationships between $CTI_{pred}$ and $CTI_{obs}$ vary depending on the warmth of a community, as indicated by the deviation from the line with the theoretical equilibrium slope of 1 (Fig. 5A). Specifically, we found a close relationship between the $CTI_{obs}$ and $CTI_{pred}$ ($F_{(1, 60)} = 1366.3$, $p < 0.001$, Fig. 5A), but the slope was significantly higher than 1 ($F_{(1, 60)} = 35.85$, $p < 0.0001$)), with cooler communities being consistently warmer than predicted. Finally, we found that communities at higher latitudes had a higher $CTI_{obs}$ compared to the $CTI_{pred}$ (latitude x CTI type = $F_{(1, 120)} = 24.36$, $p < 0.001$). Thus, communities dominated by cold-affiliated species alone (Fig. 5A) and communities at higher latitudes (Fig. 5B) were further from equilibrium compared to those with warm-affiliated species at lower latitudes.

Overall, the combination of patterns in $CTI_{pred}$ and $CTI_{obs}$ (Fig. 5) coupled with patterns in net colonisations and net extinctions in the different thermal affinity groups across bioclimate zones (Fig. 4) attests to variation in the thermophilisation pathways across latitudes. Specifically, we infer that in the south, communities are experiencing thermophilisation through diversification. The evidence is two-fold. First, there are higher net rates of colonisation in the south compared to the north. Second, warm-affiliated species and their potential colonists only occur in the south of Finland. By contrast, northern communities are undergoing thermophilisation through homogenisation, as evidenced by higher net extinctions compared to net colonisations, by a consistent deficit of potential colonist species and by the lack of any warm-affiliated species (Fig. 4). Overall, turnover emerges also as an underlying process driving thermophilisation throughout the system, as species are both leaving and joining the communities at similar net rates in each bioclimatic zone.

## Discussion

Our results revealed strong context-dependence in the pathways to thermophilisation for moth communities, where community warming is driven by the extinction of cold-affiliated species in the north, but by high colonisation rates across all thermal groups in the south. At the community level, latitudinal differences in the balance between colonisation and extinction is reflected in overall community temperature. Cold-adapted communities are currently further away from the community temperature expected at equilibrium. These deviations suggest that northern communities are currently experiencing the highest imbalance between extinctions and colonisations and that some of their species may currently be living outside of their preferred temperature range—thus creating an extinction debt[34,35]. Our approach pinpoints the mechanisms driving thermophilisation and provides empirical evidence for heightened extinction risk among cold-adapted species. Below, we discuss each finding in turn.

Overall, we found that northern communities are experiencing thermophilisation at almost double the rate of those in the south. Indeed, northern communities face higher risks under anthropogenic climate change, since they experience more rapid environmental change compared to lower latitudes[14,30]. Consistent with this expectation, we found that the rate of community warming varies with latitude, which is also consistent with previous observations of faster warming of northern communities of birds in Sweden[10] and butterflies in Canada[36]. Although the absolute rate of change in CTI was relatively small (+0.015 °C per year to +0.029 °C per year for southern and

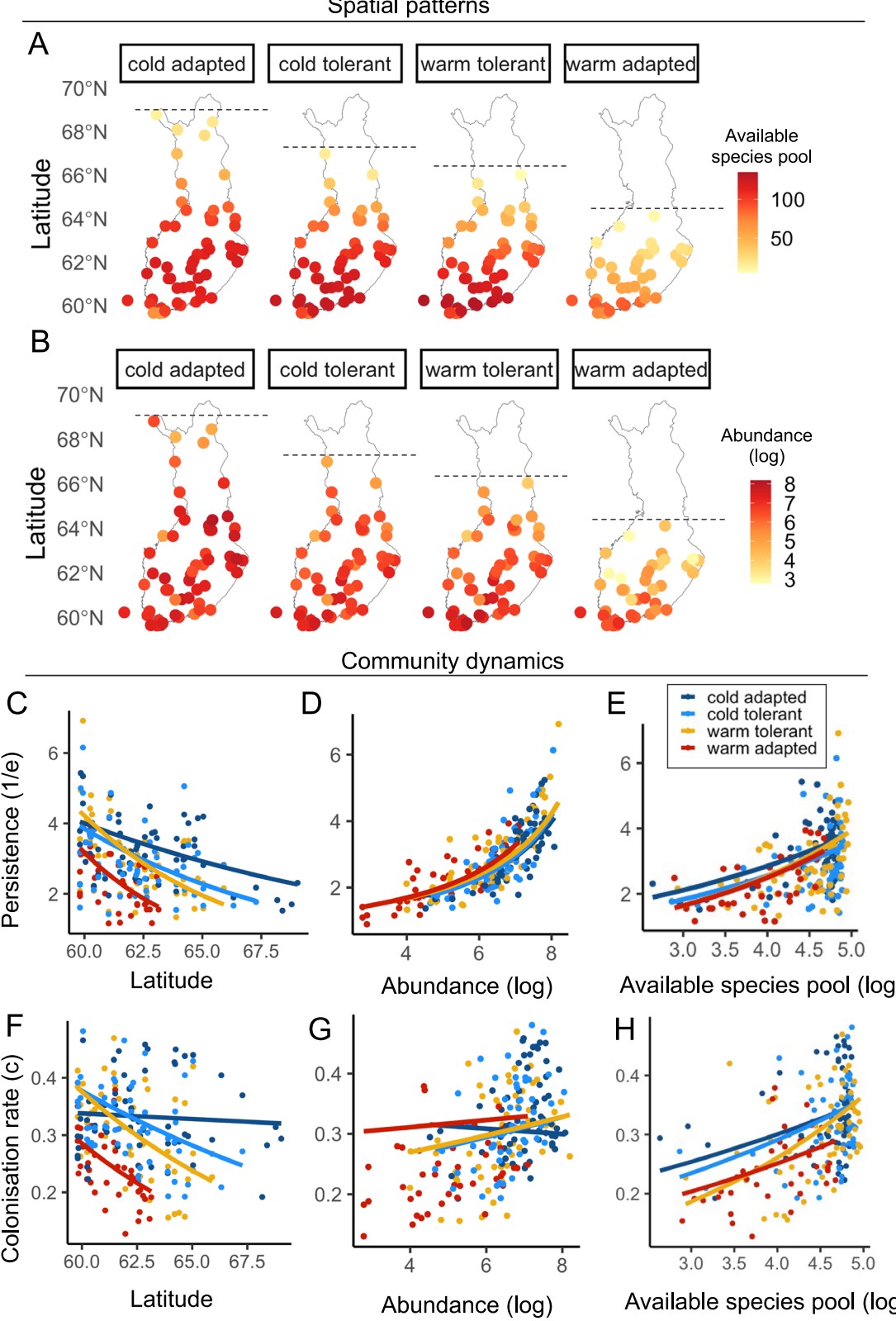

**Fig. 3 | Spatial patterns and community dynamics for the four moth thermal affinity groups.** Top panels: Maps of Finland showing the spatial distribution of each thermal affinity group, with the maximum latitude of their occurrence denoted by a dashed line and site-specific data points coloured by: **A** the size of the available species pool (i.e., the number of species which could colonise a site based on their distribution, averaged across years) and **B** the abundance of moths observed in a site, averaged across years and on a logarithmic scale. Middle panels: relationships between persistence and **C** latitude, **D** abundance, and **E** available species pool. Bottom panels: relationships between colonisation rates and **F** latitude, **G** abundance, and **H** available species pool. Each point is a site-specific observation ($n = 62$). The lines in panels **C**–**H** are coloured according to thermal groups (cold-adapted (dark blue), cold-tolerant (light blue), warm-tolerant (yellow), warm-adapted (red)) and represent the model fits using a maximum-likelihood modelling approach (see Supplementary Table 3 for full model outputs).

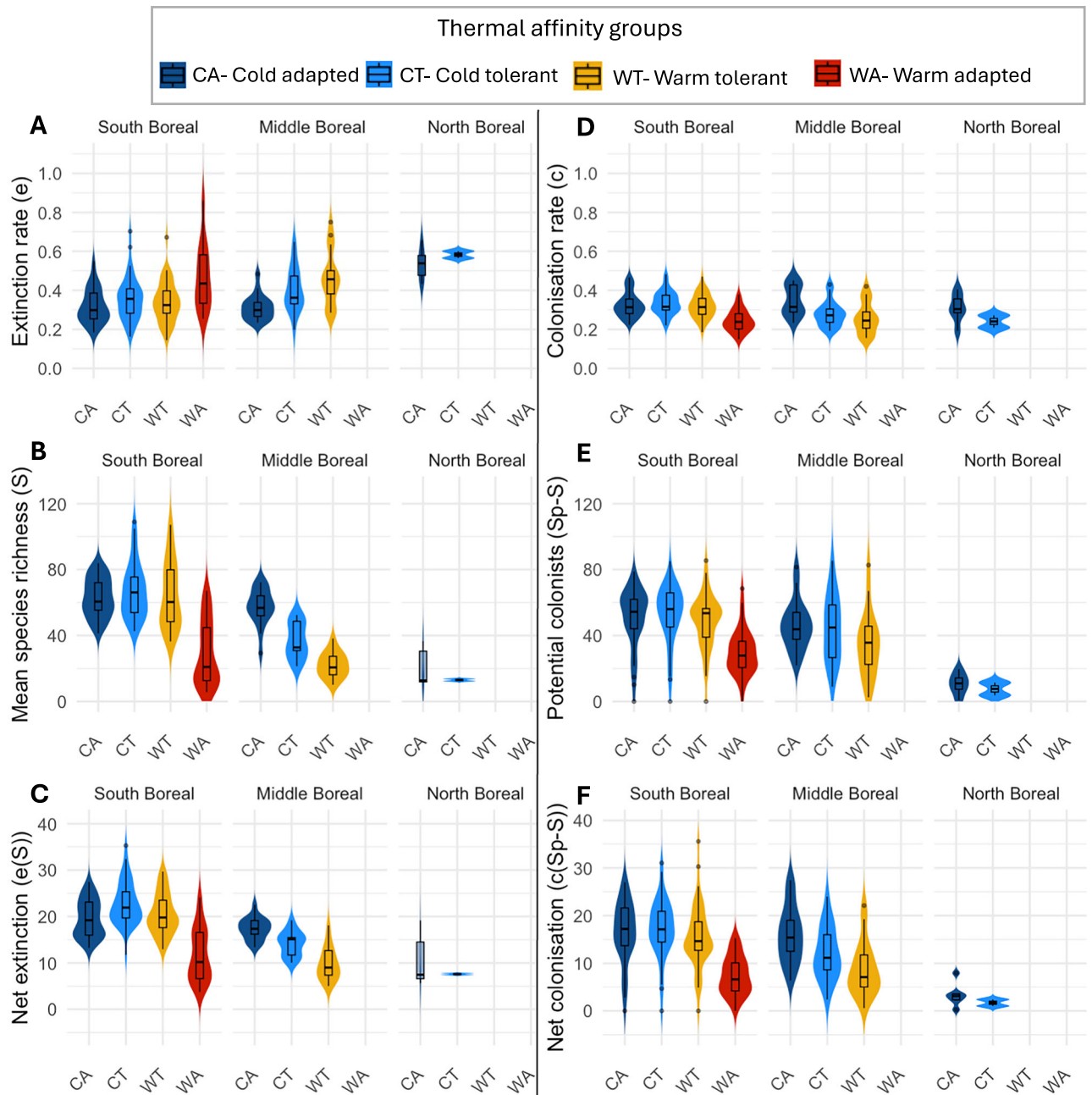

**Fig. 4 | Patterns in the components of community changes across thermal affinity groups and bioclimatic zones.** **A** Extinction rates as depicted in Fig. 2 and Fig. 3; **B** Mean species richness (total species richness per site per year, averaged across years); **C** Net extinctions (i.e., average number of species lost per year); **D** Colonisation rates as depicted in Fig. 2 and Fig. 3; **E** Potential colonists (i.e. the number of species that could colonise the target community from the surrounding region); **F** Net colonisation rates (i.e., average number of species gained per year). Each panel shows violin plots (depicting the kernel density of site-level values) overlaid with boxplots (summarising the central tendency and spread of the site-level values). The unit of analysis is the site, and sample sizes (n) per group are as follows: South Boreal: CA (n = 39), CT (n = 39), WT (n = 39), WA (n = 39); Middle Boreal: CA (n = 17), CT (n = 17), WT (n = 17); North Boreal: CA (n = 6), CT (n = 2). Thermal affinity groups are defined as: CA = cold-adapted (dark blue); CT = cold-tolerant (light blue); WT = warm-tolerante (yellow); WA = warm-adapted (red). Boxplots show the median (central line), interquartile range (IQR; bounds of the box represent the 25th and 75th percentiles), and whiskers extending to the most extreme data points within 1.5× IQR. Data points beyond this range are plotted as outliers. Violin plots display the estimated probability density of the data at different values, with wider sections indicating higher densities.

northern Finland, equalling 1.5–3 °C per century), it is lagging behind the +1 to +2 °C increase in ambient temperatures in Finland over the same period[30,37–39]. The resulting mismatch in biotic and abiotic warming is reflected in a substantial climatic debt in Finnish moth communities, whereby the mean annual change in temperature outpaces the community thermal mean (aligning with Mäkinen et al.[40]). While the ecological consequences are yet to be understood, the rift between biotic and abiotic temperatures is a clear warning sign that

the many species are drifting outside of their thermal optima with consequences for species performance and the maintenance of key species interactions[41].

We revealed the key process underpinning differential rates of community warming along the latitudinal gradient by quantifying the spatial variation in colonisation-extinction dynamics. In line with[27,32,33,42] we found that the species occurring at a given site were reflected by a dynamic balance between extinctions and colonisations.

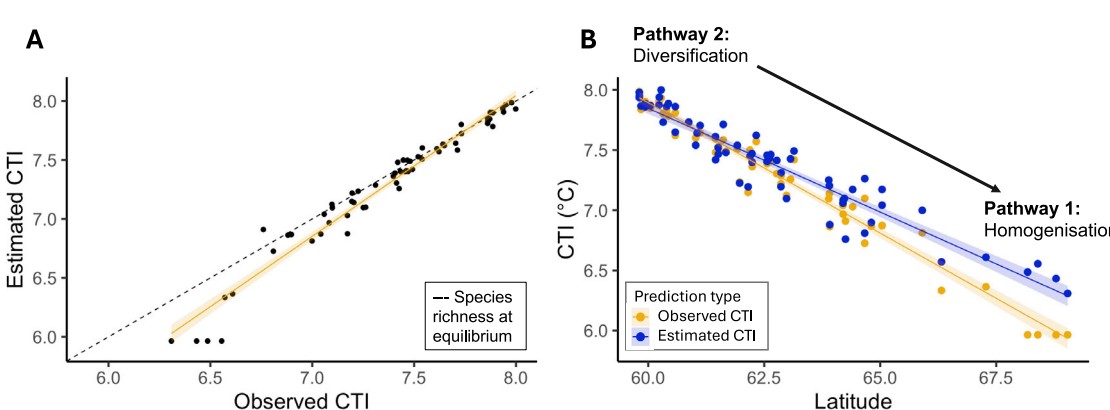

Variation in thermophilisation pathways

**Fig. 5 | Comparison of estimated and observed CTI values of moth communities along latitude. A** Relationship between the estimated community temperature index based on species richness at equilibrium (see 'Methods': Species richness at equilibrium), and the observed CTI values. The dashed line represents a 1:1 relationship, indicating equilibrium. The yellow line shows the linear model fit (mean predicted values). **B** Relationship between estimated CTI (purple) and observed CTI (yellow) of moth communities along latitude, illustrating the hypothesised thermophilisation pathways shown in Fig. 1: diversification in the south and homogenisation in the north. Each point is a site-specific observation ($n = 62$), and lines represent model predictions (mean fitted values). Shaded ribbons represent 95% confidence intervals around the fitted predictions.

We show that the size of the available species pool and the overall abundance of moths were strong predictors of colonisation-extinction dynamics[19]. As hypothesised, we found that species abundance and species pools declined with latitude[43], and these spatial patterns in species pools and abundance were matched by a similar decline in colonisation rates and persistence along the latitudinal gradient (Fig. 2).

At the community level, colonisation rates were, on average, 50% lower than extinction rates (Fig. 2), showing the main driver of thermophilisation for Finnish moth communities is through the displacement of species already present in the region compared to colonisation by new ones. This finding aligns with studies such as[25,36]. However, it contradicts the general observation that the poleward expansion of species (and thus colonisation by new species) is the main driver of thermophilisation[7,44]. As a consequence of the relationship between species pools, abundance and colonisation-extinction rates, northern communities may be at particular risk with progressing climate change, as their low persistence and low colonisation rates may limit community stability and ability to survive in the face of further climate change[27,43]. The higher local extinction rates observed in moth communities may also have broader ecological implications for higher trophic levels. Moths are vital food sources for birds, bats and other insects as both larvae and adults[45,46]. Therefore, the loss of species from local communities could result in trophic cascades through a reduction in food sources (e.g. Hochrein et al.[47]). Nonetheless, we stress that our current focus is on changes in community composition and thermal trait distributions. For example, a recent study[48] found no consistent trends in Finnish moth biomass. This shows that the extent to which the patterns uncovered here reflect changes in food availability will depend on the relative selectivity of consumers[49–51].

Importantly, our estimates of species extinctions and colonisations are a product of two features: the true disappearance of species from the local community, as well as the decline of species below a detectability threshold. Clearly, rare species will be harder to observe compared to common ones. However, two considerations attest against any major impact of detectability on our conclusions. First, our samples had high coverage[52], consistently exceeding 99% (Supplementary Table 4). Second, our results are robust to the inclusion vs. exclusion of the rarest species—we found no qualitative changes when

comparing the results with singletons included vs. removed from the data (Supplementary Tables S5 and S6)[27]. Logically, even the exclusion of singletons still leaves some level of pseudo-turnover in the community time series. Nonetheless, from a functional perspective, there will be little difference between a species hitting a true density of zero and a species going functionally extinct, by becoming rare enough to contribute little to community composition and interspecific interactions[53].

The latitudinal variation in community dynamics here observed emerges from latitudinal gradients in the baseline distribution of species with different thermal niches. Clearly, more heat-preferring species tend to be concentrated in the south of the country and more cold-preferring in the north. This was seen in different northern range limits of the different thermal affinity groups (Fig. 3). While partly a truism, this demonstrates a key impact of abiotic conditions in general —and temperatures in particular—on species membership and persistence in local communities. It also shows the importance of taking the regional and local species pools into account when evaluating patterns of colonisations and the local abundance of species when evaluating patterns of extinctions. A species can only colonise and successfully remain in a target community if it has already reached the surrounding region and reaches an abundance in which it can persist. These considerations reflect the importance of accounting for community saturation.

Temperature, however, is not the only factor limiting species occurrence. Individual species are often absent from substantial areas within their thermal niche range[15], pointing to other factors limiting their distribution, such as dispersal abilities[6], phenological mismatches[54], species interactions[55], habitat availability[56], precipitation and soil properties[15]. From this perspective, our current results are squarely focused on what part of community dynamics can be attributed to temperature, against the backdrop of all other factors. What emerges is a clear signal of temperature as such (as also reflected in abundance patterns). At the northern edges of their distribution, cold-tolerant, warm-tolerant and warm-adapted moths show low abundances and small species pools as a consequence of low persistence and low colonisation rates. Indicating that, at their range limits, species are characterised by low persistence and frequent turnover[57]. Interestingly, the cold-adapted species group showed a shallow relationship between colonisation rates and latitude (Fig. 3F). At the relatively

high latitudes of Finland, cold-adapted species are essentially facing the warm edge of their range towards the south, whereas warm-adapted species are facing their cold edge towards the north. The difference in latitudinal trends in persistence is consistent with the observation that cold range edges can track their climate change better than warm edges[15].

Finally, as a critical insight, our analytical framework allowed us to attribute the spatial variation in the rates of change in CTI to variation in colonisation-extinction dynamics, both at the level of the full community and for different thermal affinity groups. Our findings reveal that thermophilisation occurs via different pathways (Fig. 1) depending on latitude, which emerges as the net outcome of opposite forces: in the south, thermophilisation occurs through diversification (pathway 2 Fig. 1), as new species become established via high net colonisations, large species pools and higher persistence due to high abundances. In the north, thermophilisation results from the disappearance of cold-adapted species (pathway 1 Fig. 1) due to their high net extinction, low persistence and low abundances. Throughout Finland, we can see that thermophilisation is occurring through major community turnover (pathway 3 Fig. 1), with selected thermal affinity groups decreasing across bioclimatic zones (Fig. 4).

While offering a general match with the idealised thermophilisation pathways outlined in Fig. 1, our results have revealed some important nuances. In the south, colonisation occurred more frequently than in the north as expected. Yet, contrary to expectations, warm-adapted species exhibited the lowest net colonisation rates (Fig. 4F). This suggests that thermophilisation in the south is driven primarily by the persistence of cold-affiliated species and colonisation of warm-tolerant species whereas colonisation by warm-adapted species appears to be a secondary driver. Nonetheless, in the south, the warm-adapted group had the highest fraction of potential colonists in relation to their local species richness; these pools were nearly one-third greater than locally observed species richness. The relatively large pools of potential colonists indicate that colonisations are likely to play a significant role in future community thermophilisation under continuing climate warming.

In the north, net extinctions exceeded net colonisations, as we expected. This indicates that thermophilisation via homogenisation is due to the extinction of the coldest-adapted species within the cold-adapted group; however, our thermal grouping approach limits our inference. Although using thermal groupings is necessary for broad-scale analysis, such grouping results in a loss of resolution regarding which species in particular are being lost or persist. For example, within the cold-adapted thermal group, there is considerable variation in the STIs (see Fig. 4), with some species being highly specialised to cold extremes and others being more generalist. Such heterogeneity within our thermal affinity groups may be driving some deviations from our hypothesised patterns.

Overall, we had expected cold-adapted species to experience extinction in the south and persistence in the north, with warm-adapted species persisting in the south and colonising the north. However, our results suggest a more complex interplay of colonisation and extinction dynamics, shaped by local species pools, community abundances and regional thermal constraints, as shown in marine systems by Moore et al. [15]. Here, our focus on a single country—as forming a subset of most species' ranges—will hide some of the ongoing processes. Once data become available, future research should extend beyond country-level analyses to incorporate global or continent-wide assessments of thermal groupings. A pan-European perspective may reveal species movements across latitudinal gradients and better resolve the drivers of thermophilisation.

To conclude, our paper outlines a community-level approach to understanding how colonisation-extinction dynamics drive thermophilisation, and shows how this varies in geographic and environmental space. We showed that rapid thermophilisation of moth communities was associated with the loss of cold species in particular. Compounding the plight of northern communities are lower colonisation rates with smaller species pools and higher local extinction rates with decreasing abundances. Beyond resolving the population-level mechanisms behind community warming, our findings highlight the vulnerability of species and communities at their range limits, while also showing an increasing rift between community composition and the surrounding climate, resulting in accumulating debts as climate warming progresses.

## Methods

We used a dataset of systematically sampled moth communities in a country-wide monitoring scheme in Finland. Our data covers 30 years between 1993–2022 and consists of 658 species of macro moths sampled across 62 sites spanning a latitudinal gradient of 1200 km (Supplementary Fig. 1). To understand whether communities are warming in response to climate change, we characterised the species in our observed communities by their thermal affinity i.e. the species temperature index, STI[8], which is measured as the average temperature recorded in their European range. We then calculated the community temperature index CTI, which is the average STI weighted by species abundances at each site. We first analysed changes in CTI over time and across latitudes. Then, applying a dynamic stochastic modelling approach, we characterised community colonisation-extinction dynamics and revealed their impacts on community thermophilisation along this latitudinal gradient. All data manipulation and analysis were done using R version 4.4.3 unless otherwise stated[58]. Please see Supplementary Table 7 for all R packages used.

### Study system and dataset

Finland spans from 59° to 70° degrees North, thus encompassing diverse climatic zones (Supplementary Fig. 2A), from temperate maritime (hemi-boreal) to Arctic (northern boreal). This renders the country ideal for investigating latitudinal variations in species distributions and community dynamics (Supplementary Fig. 1 and Fig. 2A).

Moth data were obtained from the Finnish National Moth Monitoring Scheme (Nocturna)[59,60], and were extracted from the moth monitoring database (Yöpeti) on 18 December, 2023. Moth communities were sampled using light traps (model 'Jalas', with 160W mixed light or 125W Hg vapour bulbs). Light traps were emptied weekly between early spring and late autumn yearly between 1993–2022 and individuals were identified to the species level by volunteer experts. In the current study, we focused on macro moths (Supplementary Table 8), since micro moths had been inconsistently scored from the samples. To ensure a robust and unbiased dataset for our analysis, we filtered the data for obvious sources of error. Specifically, (i) to confine our data to sites for which temporal change can be reliably established, we chose sites with at least 10 years of data; and (ii) to derive a comparable baseline between sites, we excluded any sites that started data collection later than 2005. These filtering steps resulted in 728 species in 62 sites. We then used the 'iNEXT' package (version 3.0.1,[52,61]) `iNEXT::iNEXT` to evaluate whether our sites exhibited sufficient coverage of the focal communities. Based on sample completeness using rarefaction[52,61], all sites showed a sample-based coverage of >99% (Supplementary Table 4).

Since rare species are likely to be associated with low detection probabilities (and thus with high rates of pseudo-turnover[27], we explored whether our analysis was affected by the inclusion vs. exclusion of rare species. To this aim, we ran the analysis (see section Model fitting and inference below) with singletons (i.e. species observed as single individuals) included vs. removed from the dataset. We found that estimates of both colonisation rates and extinction rates were highly correlated between the two data sets ($r = 0.82$–$0.88$, respectively). As there were no qualitative differences in results

between the two sets, we report the results with singletons included in the main text, and results with singletons removed in Supplementary Tables S5 and S6.

## Calculating species (STI) and community temperature index (CTI)

Species climatic niche metrics were calculated following[6,8]. The workflow was as follows: first, species distributions were extracted from European Atlases[62–81]. The atlas maps were then digitised and georeferenced with the CGRS grid (Common European Chorological Grid Reference System from the European Environment Agency[82]). We obtained distribution maps for 658 species from the 728 species in the filtered dataset. Climate data for the corresponding grid was interpolated using methods developed in the ALARM project (following[83,84]). Based on species occupancy of grid cells across Europe, climatic niche metrics were calculated by overlaying range maps with climatic variables over the period between 1961 and 1990, which is the period just before our sampling started. Since this is also the period before accelerating climate change sets in, it represents relevant baseline conditions[85].

To characterise the climatic niche of a species, we used the standard definition of STI−i.e. the average mean temperature across its range[8]. While a similar metric could be derived for other climatic descriptors, we found mean temperature offered a good proxy for a range of metrics (as shown in Schweiger et al. [8]). Across Europe, mean temperature shows high correlation ($r > 88\%$) with a range of other temperature measures (Supplementary Fig. 3).

For the community-level climatic niche, we calculated the community temperature index (CTI) for each site in each year by weighting each species' STI value by its abundance before calculating the weighted mean.

## Community temperature index change

To test for changes in the CTI through time and across latitude, we used linear mixed models (lme4::lmer[86]) with the site as a random effect. The following model structure was fitted:

$$\text{CTI}(^{\circ}\text{C}) \sim \text{Year*Latitude} + (1|\text{Site}) \qquad (1)$$

To capture site-specific changes in CTI through time, we fitted a linear model (car::lm[87]) with site as a fixed effect instead of latitude and used the site-specific slopes for visualising the spatial changes of CTI (Fig. 2A).

All model fits and terms were statistically tested using analysis of deviance type II Wald chi-square tests.

## Measuring community dynamics using colonisation-extinction rates

Our approach to understanding community dynamics is based in MacArthur and Wilson's Theory of Island Biogeography[32], as implemented in a basic stochastic colonisation-extinction model developed by Alonso et al. [27]. Although this is a relatively simple approach, these models can achieve good predictive power when examining how communities reassemble under disturbance[27,42,88]. The approach assumes that the state of a site's community can be described by the presence or absence of each species each year. We can then measure how the state of the site changes over time due to random colonisation and extinction processes. Each species is assumed to randomly colonise unoccupied sites at a rate $c$ and to go extinct in occupied sites at a rate $e$. Under MacArthur and Wilson's[32] assumption of species equivalence and independence, all species have a common $c$ and $e$.

As occupied sites undergo extinctions at a rate of $e$, the average time to go extinct is $1/e$. Here, we focused on this quantity ($1/e$), which we henceforth refer to as "persistence"[89]. Focusing on persistence avoids any undesired connotations of extinction rate, which may be

read as referring to regional extinction rather than transitions from presence to absence at a single site.

## Biotic community measurements

It has been shown that colonisation is largely driven by regional richness (species pools) and non-specific local extinctions can be attributed to community assembly features, such as the initial state of the community, abundance and diversity[32,33]. Therefore, we related colonisation and extinction rates to the overall community abundance of each site and to the size of the available species pools[90]. We calculated overall community abundance as the total number of individuals at each site, which was averaged across years. The regional species pool of a site was defined as the total set of species that could potentially occur at the site, averaged across years. We used occurrence data for all Finnish moths as derived from our dataset to define the maximum and minimum latitudes of occurrence. We included species in the regional pool if their distribution overlapped with the site's location (Supplementary Fig. 5).

## Model fitting and inference

Following the reasoning above (see Measuring community dynamics using colonisation-extinction rates), we used R package 'island' (version 0.2.10,[28]) to estimate the extinction and the colonisation parameters. The analysis was performed in two steps: first, we estimated colonisation and extinction rates for each site or each thermal group at each site for visualisation purposes (function `island::irregular_sampling_scheme`), which are the $c$s and $e$s plotted as points in Figs. 2 and 3.

Secondly, to test how abiotic and biotic variables influence community dynamics, we assumed that colonisation rates are a simple exponential function of some explanatory variable:

$$c = \exp(\beta_0 + \beta_1 * X) \qquad (2)$$

where $X$ is the explanatory variable and the $\beta$'s are the estimated parameters. The same function was assumed for extinction rates ($e$). As the $c$'s and $e$'s were positive, we used exponential functions to ensure the fitted values were also positive. We used the function `island::NLL_isd`, which returns the negative log-likelihood of a pair of colonisation and extinction rates for each of our 62 sites and wrote a function that sums these across all sites. We then used the 'bbmle' package (version 1.0.25.1) `bbmle::mle2` to estimate the parameters using maximum likelihood[91]. As explanatory variables $X$ we used latitude, mean abundance, and available species pool. As these variables were somewhat co-linear (between 57-97%; Supplementary Fig. 7), we fitted separate models for each.

**Thermal affinity groups.** To address how thermal niches influence patterns of community warming, we relaxed the original assumption of species equivalence (see *Measuring community dynamics using colonisation-extinction rates*) and grouped species into thermal affinity groups. We split species into four groups based on their thermal affinities and estimated the colonisation-extinction parameters for each group. We divided the STI values into quartiles, thereby defining four groups: (1) cold-adapted species, (2) cold-tolerant species, (3) warm-tolerant species, and (4) warm-adapted species (Supplementary Fig. 4). Colonisation and extinction rates were then generated for each thermal group at each site. Colonisation and extinction rates were fitted to the explanatory variables using the maximum-likelihood approach as above, generating parameters for each group (i.e. each with a unique intercept and slope).

**Species richness at equilibrium.** Quantifying extinction and colonisation rates enables us to understand how communities diverge from their equilibrium species richness[33]. A community at equilibrium can

be defined when $c(Sp\text{-}S) = e(S)$[27], where $c$ is the colonisation rate, $e$ is the extinction rate, $Sp$ is the available species pool, and $S$ is the average species richness observed in the site.

Based on this balance between extinction rates and colonisation rates, we derived the expected species richness of each group $i$ at each site $j$, which is given by

$$\text{Estimated}_{\text{species richness},i,j} = \text{Species pool}_{i,j}\left(\frac{c_{i,j}}{e_{i,j}+c_{i,j}}\right) \quad (3)$$

Here, $Speciespool_{i,j}$ is the species pool of thermal affinity group $i$ at site $j$, and $c_{i,j}$ and $e_{i,j}$ are the estimated colonisation and extinction rate of thermal group $i$ at site $j$[27].

Then, using the estimated species richness of each thermal group, we calculated the expected CTI ($CTI_{\text{pred}}$) of site $j$ as

$$CTI_{\text{pred},j} = \sum_{i=1}^{4}\left(\frac{\text{Estimated}_{\text{species richness},i,j}{}^{*}\overline{STI}_{i,j}}{\sum_{i}\text{Estimated}_{\text{species richness},i,j}}\right) \quad (4)$$

where $Estimated_{speciesrichness,i,j}$ is the estimated species richness for thermal group $i$ at site $j$. $\overline{STI}_{i,j}$ is the average $STI$ of thermal group $i$ at site $j$. We then compared $CTI_{pred}$ with our observed CTI ($CTI_{obs}$), which was calculated in the same equation as above (3) but replaced $Estimated_{speciesrichness,i,j}$ with $Observed_{speciesrichness,i,j}$, which are the recorded species richness of each thermal group $i$ at each site $j$.

Using linear models (lm::car), we fitted $CTI_{\text{obs}}$ as a function of $CTI_{\text{pred}}$, and statistically tested their relationship (car::Anova). If communities are at equilibrium, then the $CTI_{\text{obs}} \sim CTI_{\text{pred}}$ relationship should have a slope of 1. We thus tested if the model fit was significantly different from a slope of 1 by incorporating an offset to test the deviation from the hypothesised slope. Finally, we examined how $CTI_{\text{pred}}$ and $CTI_{\text{obs}}$ varied along latitude by fitting a linear model

$$CTI\left(^{\circ}C\right) \sim \text{Latitude*Prediction type} \quad (5)$$

where CTI is either $CTI_{pred}$ or $CTI_{obs}$ (prediction type). If the interaction $p < 0.05$ it indicates that the thermophilisation pathways vary with latitude.

### Reporting summary

Further information on research design is available in the Nature Portfolio Reporting Summary linked to this article.

## Data availability

The raw moth data used in this study are openly available from the Finnish Environment Institute, Finnish National Moth Monitoring, Biodiversity Information Facility at https://laji.fi/en/observation/list?collectionId=HR.4511. All other data, including raw species temperature niche data and cleaned moth data used in this study, are available in the Dryad database: https://doi.org/10.5061/dryad.qbzkh18s7.

## Code availability

All code used in this manuscript is deposited in Dryad: https://doi.org/10.5061/dryad.qbzkh18s7.

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

## Acknowledgements
We thank Ruby Fries and Elisa Hanhirova, who digitised many of the moth range maps. We would like to thank Pinja Kettunen for designing the excellent moth images in Fig. 1. We also thank all the volunteers over the last 30 years who have sampled and identified the moths in our dataset. The moth monitoring scheme has been supported by the Finnish Ministry of the Environment. This project was funded by the Jane and Aatos Erkko Foundation. TR was funded by the European Research Council (ERC) under the European Union's Horizon 2020 research and innovation programme (grant agreement No 856506: ERC-synergy project LIFEPLAN) and by the Swedish Research Council Vetenskapsrådet, Decision 2023-05118. LHA was funded by the Research Council of Finland (grants 340280 and 361416).

## Author contributions
E.E.E., L.H.A., A.D., J.M., M.R. and T.R. conceived the idea. E.E.E., M.R., J.V. and T.R. analysed the data. I.C., I.-M.H., R.L., J.P. and A.S. curated, collected and provided the moth trapping data. M.S., A.-L.L. provided feedback on the project. E.E.E. wrote the first draft of the manuscript, and all authors read, edited, and approved subsequent revisions.

## Competing interests
The authors declare no competing interests.
