## [Transparent Peer Review file · Nature Communications]

Recent community warming of moths in Finland is driven by extinction in the north and colonisation in the south

Corresponding Author: Dr Emilie Ellis

Version 0:

Reviewer comments:

Reviewer #1

(Remarks to the Author)

Dear Editor and Authors,

Thank you for the opportunity to review your manuscript entitled “Extinction in the north and colonization in the south: the latitudinal drivers of community warming” submitted for consideration to Nature Communications. This topic is of particular interest to me as someone who works both in Lepidoptera and in the boreal/Arctic interface. Further, seeing as a manuscript covers typically under-sampled moth biodiversity, I believe this contribution to the literature is exceedingly valuable. Decomposition of thermophilization into constituent processes is also of great value for predicting patterns of Lepidopteran diversity under global change.

All said, I am happy to recommend this manuscript for minor revisions pending the authors’ responses to my comments below:

1. Out of curiosity, I wonder if STI are correlated at all with any other biologically/ecologically relevant traits that may also relate to extinction and colonization rates. For example, you might expect wingspan to be correlated with STI, with warmer STI species having higher wingspans due to longer growing seasons/windows to acquire host plant material. This in turn could impact colonization rates. Not something you need to do an analysis on but worth considering.
2. Figure 3 is stunning.
3. In the discussion, I wonder if the authors can opine a bit more on the consequences of these dynamics on ecosystem/community-level processes. Moth caterpillars are an abundant food source for other organisms and the elevated extinction rates in the north may be problematic, especially for migratory bird species for example. I appreciate the author’s brevity in the overall writing of the manuscript but wish there were more connections to the ecological system as a whole.

I want to reiterate again that this paper was a pleasure to read and no doubt represents the high-quality work that comes out of Finland regarding global change dynamics. I look forward to reading a future revision and additional work from the authors.

Reviewer #2

(Remarks to the Author)

The manuscript “Extinction in the north and colonisation in the south: the latitudinal drivers of community warming” by Ellis et al. finds that thermophilisation is faster at higher latitudes in Finland for moth communities. This latitudinal gradient of thermophilisation is shown to be the outcome of differential colonization and extinction of warm and cold affiliated species. It adds an interesting twist to the current literature, identifying the “pathways” of thermophilisation using island biogeography models.

The methodology is sound, makes a creative use of stochastic island biogeography models to predict thermophilisation, and allows for replication of the results. However, I have some concerns with the interpretation of some results, which I outline below.

1. Pathway identification.

My main concern with the manuscript lies in the interpretation of Figure 4B. It is difficult to see how you assign a pathway for

the different latitudes. Probably, it is done taking into account colonization and extinction rates with latitude, but I fail to see an explicit explanation of your interpretation, making it difficult to fully grasp what is going on with latitude.

If I understood correctly the different pathways of thermophilisation, I would suggest the addition of a bar graph showing the deviation of the observed richness from the expectation across species groups, along a ranking of latitude. The graph would indicate if there's a deficit/excess of cold species and a higher number of warmer ones, compared to the expectation at equilibrium. This would link with the conceptual figure 1 and facilitate the understanding of Figure 4B.

Related to this, in l.225-226, I fail to see how a large variation in colonization extinction dynamics allows for the inference of turnover causing thermophilisation there. The figure I outlined above may help showing which groups end up having higher numbers than expected.

Other comments

2.
In l. 148, I do not understand how your results suggest an extinction debt in the north, as fig. 2 shows the high slopes of CTI in the north, and the sites there are assigned to the pathway of homogenization, which implies extinction of cold species.

3.
In l. 304-305 you talk about the assumption of neutrality of MacArthur and Wilson (1967). I doubt that they made such a claim. Their graphical model (in Figure 7 of the book) implies equivalence (in the sense that species identities are interchangeable for modeling purposes) but not independence (curved lines in Fig 7 imply a dependence of colonization and extinction rates with the number of species present, that is, species interactions). Although this can be considered neutrality in some sense (I do not fully agree, as their model implied species interactions even if we do not care of their identities), it does not take into account the ultimate goal of MacArthur and Wilson of obtaining a "biogeography of the species" (as they call it in their prospect, chapter 8, and Wilson acknowledged in Losos and Ricklefs book). That being said, it is true that Alonso et al. 2015 and Ontiveros et al. 2019 assume species equivalence and independence, as a mathematical tool to facilitate parameter estimation (as Simberloff 1969 did to facilitate the mathematical treatment of the island biogeography model). So, I would say here just "under the assumption of species equivalence and independence, all species ...".

I encourage the authors to address these comments and improve their already interesting and sound manuscript.

Reviewer #3

(Remarks to the Author)

This contribution uses 30 years of macromoth collecting and reports a poleward shift of warm-adapted species and extinction of cold-adapted species in the north. For this process authors apply the concept of thermophilisation, prior mainly used in plant ecology to describe alpine gradients. Technically, the paper is well written and the methods are easy to understand. The topic fits into the scope of Nat. Comm.

Numerous prior studies have reported a poleward shift in various organisms, also in Lepidoptera. Consequently, southern populations are necessarily composed of relatively warm-loving species. Authors do not clearly explain why they use the thermal detour if they might simply report species poleward shifts. The respective reassembly of communities according to thermal adaptation is evident.

A new aspect might be the selective extinction. Some species appear to be unable to adapt either to warmer temperatures or to altered resource availability. The study focuses on temperature only. However, higher temperatures alter resources (often host plants) and moth and host plant phenologies, sometimes leading to phenological mismatches. Both topics are not discussed. I think these points should be included at least in the discussion.

The present submission focuses on thermal niches, the range of temperatures a species occurs, and associated community wide temperature indices. Temperature range was calculated from the average temperatures across the occurrence range. This is a very raw approximation as mean temperature is not as decisive for Lepidoptera occurrence as, for instance, minimum temperatures and temperatures during activity periods. I wondered why the study did not use these more meaningful measures. Further, temperatures increased during the last 30 years. What was the baseline for the STI calculation? Many species are known to have adapted to new temperature regimes for instance changing their activity periods. How was this included? I think the description and the rationale behind the STI assessment needs additional information.

The description of the sampling needs clarification. In line 264 Macrolepidoptera were mentioned, while the main text speaks of Macromoths. Were butterflies included? The exclusion of singletons is unclear. Given the observed abundance gradients, particularly singletons might be of interest and an exclusion might bias the results. In line 284 I learned that 728 species (out of a total of 1755 species) were used. Does this mean that more than 50% of species were excluded? Or do the 1755 species refer to all Lepidoptera? In that case how many macromoths were collected and how many do occur in Finland? It might be that the cold adapted species did not die out but simply attained lower abundances due to resource availability.

I missed information about sample coverage. How many species are missing in the monitoring collections? What about richness estimation for particular sites (rarefaction curves)? Did the different regions have comparable sample coverages? I missed the raw data and respective summary tables for the single study regions (or traps?). Otherwise it is difficult to assess whether the results might stem from sample biases. I guess that the overall lower abundances in northern regions

also affect detection probabilities. Was there a latitudinal gradient in coverage and the frequency of singletons? In conclusion, the description of the data acquisition is insufficient.

Finally, given the high species exclusion rates and possible detection issues I wondered whether extinction and colonisation times can be calculated reliably. These issues should be clarified.

Version 1:

Reviewer comments:

Reviewer #1

(Remarks to the Author)

Dear Authors, many thanks for the opportunity to review this work again. I look forward to reading more about this forthcoming trait-based analysis in the future. Happy to recommend for publication!

Reviewer #2

(Remarks to the Author)

The authors have answered most of my concerns satisfactorily, but there are a few points where the reasoning appears to be inconsistent or even contradictory, which have emerged from their response.

1. The definition of the deviation between expected and observed richness in l. 131 contradicts the explanation in l.135-137 and Fig. 4A. If you define the deviation as $\log(\text{expected}/\text{observed})$, positive values imply a deficit of species (taking as a reference the expectation). To be perfectly clear, you can only get positive values if $\text{expected} > \text{observed}$, which I would call a deficit of species, and it is indicated as an excess of species in the text. Conversely, negative values would correspond to excess of species, while in the text it is indicated as a deficit. This needs to be clarified, because the interpretation of Fig 4a greatly varies if you interpret the results as a deficit or as an excess. Probably you just meant to define the deviation as $\log(\text{observed}/\text{expected})$.

2. If the interpretation is maintained in the same sense as it is portrayed in the text, I think it is not accurate to describe the thermophilisation pathway of the mid biozone as turnover. Turnover implies the colonization of warm species and extinction of cold ones, but Fig. 4A indicates that there is an excess of cold ones over expectation and a huge deficit of warm ones, which is opposite to the turnover pathway. I guess that it is still turnover, but not the one explained in Figure 1 (pathway 3). The interpretation of the other zones holds - in the south, you maintain (even have a few more) cold species and get some new warm ones, which is compatible with diversification, while in the north you have a deficit of cold species and maintain (no) warm ones, which is compatible with homogeneisation. Can you explain the mismatch in the mid biozone?

3. I am not really sure that the pathways, taken together and in the light of Fig 4, follow a logical succession. You could expect different pathways in different zones, sure, but getting diversification-turnover-homogeneisation in a latitudinal gradient does not seem intuitive. I would expect cold species to get extinct in the south and persist in the north, and warm species persist in the south and colonize the north. This would imply a homogeneisation-turnover-diversification sequence instead of the one shown currently in the article. Maybe you could comment on this and stress in the conclusions that the pathways you find are not the ones that may be expected a priori, if you deem it fit (it is not mandatory, though).

I encourage the authors to solve these discrepancies (or at least acknowledge them) in order to have a rounder manuscript. I also thank them for their previous responses.

Reviewer #3

(Remarks to the Author)

I was impressed by this revision. The authors fully answered to my concerns and questions and performed a number of reevaluations and additional analyses. They included the requested clarifications and extended particularly the methods section. I have no more comments.

Kid regards
Werner Ulrich

Version 2:

Reviewer comments:

Reviewer #2

(Remarks to the Author)

Thank you for your thorough revision. The pathway identification is now clear, addressing my initial concern, and everything checks out. I appreciate your excellent work!

Response to reviewer comments: *Extinction in the north and colonisation in the south: the latitudinal drivers of community warming.*

Reviewer 1

Dear Editor and Authors, Thank you for the opportunity to review your manuscript entitled “Extinction in the north and colonization in the south: the latitudinal drivers of community warming” submitted for consideration to Nature Communications. This topic is of particular interest to me as someone who works both in Lepidoptera and in the boreal/Arctic interface. Further, seeing as a manuscript covers typically under-sampled moth biodiversity, I believe this contribution to the literature is exceedingly valuable. Decomposition of thermophilization into constituent processes is also of great value for predicting patterns of Lepidopteran diversity under global change. All said, I am happy to recommend this manuscript for minor revisions pending the authors’ responses to my comments below:

Reply 1.1:

We would like to thank the Reviewer for their positive review and helpful comments, which we address point-by-point below.

1. Out of curiosity, I wonder if STI are correlated at all with any other biologically/ecologically relevant traits that may also relate to extinction and colonization rates. For example, you might expect wingspan to be correlated with STI, with warmer STI species having higher wingspans due to longer growing seasons/windows to acquire host plant material. This in turn could impact colonization rates. Not something you need to do an analysis on but worth considering.

Reply 1.2:

After conducting the current study, we have been curious about the very same question – and found that it has yet to be addressed in the literature. However, addressing it will clearly call for a separate study, as it requires the compilation of trait values for 700 species and an entirely separate set of analysis. However, we have just embarked on a new project, where we will address the relationship between STI and thermal traits (e.g. body size and wing darkness). While the project is still at a preliminary stage, we can reveal that preliminary analysis suggests STIs to be correlated with wing darkness. Our current goal is to address the very question posed by the Reviewer. In doing so, we will also learn whether colonisation-extinction patterns are filtering out certain traits. Thus, we are grateful to the Reviewer for raising this interesting point and will be keen to return with a more conclusive answer before long.

2. Figure 3 is stunning.

Reply 1.3:

Thank you!

3. In the discussion, I wonder if the authors can opine a bit more on the consequences of these dynamics on ecosystem/community-level processes. Moth caterpillars are an abundant food source for other organisms and the elevated extinction rates in the

north may be problematic, especially for migratory bird species for example. I appreciate the author's brevity in the overall writing of the manuscript but wish there were more connections to the ecological system as a whole.

Reply 1.4:

We agree the broader ecological consequences are important to discuss, and have thus added further reflections on the ecosystem/community-level consequences of the processes observed (Lines 206-213, with further consideration of Reviewer 3's comments in Lines; see **Reply 3.3** below). However, with respect to the specific perspective of food availability, we have remained careful about staying within the scope of our current evidence. Our current analysis is explicitly focused on extinction vs colonisation processes, whereas we include no analysis of overall food availability – as perhaps best measured by overall biomass. Overall changes in biomass have been addressed in a separate study (<https://doi.org/10.1111/icad.12657>, including some of the current coauthors). These authors found no consistent trends in overall moth biomass – a fact which we now mentioned in the extended discussion (Lines 206-213)

Lines 206-213: The high local extinction rates observed in moth communities may also have broader ecological implications for higher trophic levels. Moths are vital food sources for birds, bats and other insects as both larvae and adults (Evans et al., 2024; Arrizabalaga-Escudero et al., 2015). Thus, the loss of species from local communities could result in trophic cascades through a reduction in food sources (e.g. Hochrein et al. (2022)). Nonetheless, we stress that our current focus is on changes in community composition and trait distributions. E.g., a recent study (Yazdanian et al., 2023) found no consistent trends in Finnish moth biomass. Thus, how the patterns uncovered here might reflect into changes in food availability will depend on the relative selectivity of consumers (Vesterinen et al., 2020; Kaunisto et al., 2020; Vesterinen et al., 2018)

I want to reiterate again that this paper was a pleasure to read and no doubt represents

the high-quality work that comes out of Finland regarding global change dynamics. I look forward to reading a future revision and additional work from the authors.

Reply 1.5:

Thank you again for your encouraging review!

Reviewer 2

The manuscript “Extinction in the north and colonisation in the south: the latitudinal drivers of community warming” by Ellis et al. finds that thermophilisation is faster at higher latitudes in Finland for moth communities. This latitudinal gradient of thermophilisation is shown to be the outcome of differential colonization and extinction of warm and cold affiliated species. It adds an interesting twist to the current literature, identifying the “pathways” of thermophilisation using island biogeography models.

The methodology is sound, makes a creative use of stochastic island biogeography models to predict thermophilisation, and allows for replication of the results. However, I have some concerns with the interpretation of some results, which I outline below.

Reply 2.1:

We thank the Reviewer for their positive and constructive review and are particularly grateful to them for suggesting ways that we clarify how we inferred differences in pathways to thermophilisation across latitudes.

1. Pathway identification. My main concern with the manuscript lies in the interpretation of Figure 4B. It is difficult to see how you assign a pathway for the different latitudes. Probably, it is done taking into account colonization and extinction rates with latitude, but I fail to see an explicit explanation of your interpretation, making it difficult to fully grasp what is going on with latitude.

Reply 2.2:

As this is the key point of our analyses, we have now done our utmost to clarify the inference adopted. The following explanation has been added to Lines 125-146 and Lines 156-163), along with the figure suggested by the Reviewer (see **Reply 2.3**).

Lines 131-146: As a metric of deviations between the expected and observed richness, we use $\log(\text{expected species richness}/\text{observed species richness})$, noting that an excess

of species will be observed when $c(Sp-S) < e(S)$, where c is the colonisation rate, Sp is the available species pool, S is the species richness observed at the site and e is the extinction rate. Thus, a deficit of species will be observed when $c(Sp-S) < e(S)$. Comparing the expected-to-observed species richness (Figure 4A), a value of zero will indicate equilibrium ($c(Sp-S) = e(S)$), a positive value will indicate an excess of species and a value lower than zero indicates a deficit of species. We found mixed patterns across thermal affinity groups and latitude. Specifically, in the south there was an excess of species for all groups except the warm-adapted moths (Figure 4A). At mid-latitudes, there were both deficits and excesses in thermal affinity groups, with the largest deficits in warm-adapted moths. At high latitudes there were deficits in cold-adapted and cold-tolerant species (Figure 4A). We note that no warm-tolerant or warm-adapted species reach this far north (Figure 3)

Lines 156-163: Overall, the combined pattern between CTIpred and CTIobs (Figure 4B, 4C) coupled with the patterns of thermal affinity group species richness estimates (Figure 4A) both attest to variation in the thermophilisation pathways across latitudes (shown in Figure 4C). We infer that in the south, communities are experiencing thermophilisation through diversification, as evidenced by an excess of species. At mid-latitudes, communities are experiencing thermophilisation through turnover, given the mix of species deficits and excesses among thermal affinity groups. Finally, in the north, communities are undergoing thermophilisation through homogenisation, as evidenced by consistent a deficit of species and currently warmer communities than expected.

If I understood correctly the different pathways of thermophilisation, I would suggest the addition of a bar graph showing the deviation of the observed richness from the expectation across species groups, along a ranking of latitude. The graph would indicate if there's a deficit/excess of cold species and a higher number of warmer ones, compared to the expectation at equilibrium. This would link with the conceptual figure 1 and facilitate the understanding of Figure 4B.

Reply 2.3:

We thank the reviewer for this excellent suggestion. We have now added a new panel in figure 4 (Figure 4A), showing that in the southern bioclimatic zone, species richness is higher than expected under long-term equilibrium. This suggests that this zone is currently characterised by high rates of colonisations (diversification). The middle, boreal zone shows a mix between lower and higher species richness than expected, depending on the thermal group considered, suggesting turnover in community composition. By comparison, the northern zone is characterised by lower species richness than expected under long-term equilibrium. This suggests homogenisation of the fauna, by the selective loss of cold-adapted and tolerant species. This rationale is now clarified in the main text (with details in **Reply 2.2** above and with the new text found on Lines 125-146 and Lines 156-163 and in the methods: Lines 395-397).

Related to this, in l.225-226, I fail to see how a large variation in colonization extinction dynamics allows for the inference of turnover causing thermophilisation there. The figure I outlined above may help showing which groups end up having higher numbers than expected.

Reply 2.4:

We apologise for this misunderstanding, which was caused by our poor wording. The word “variation” was intended to refer to “differences between thermal groups”, which should suffice to explain our inference. To clarify our intended meaning, we have now changed the phrasing of Lines 251-252 to say that extinction and colonisation are both high but in different thermal groups, which results in turnover. When combined with the other clarifications offered above (**see Reply, 2.2 and 2.3**), we hope that our inference has now been clarified throughout the MS.

Lines 251-252: **In the middle of Finland, we see thermophilisation through high**

turnover, with some groups decreasing and others increasing in dominance

Other comments 2. In l. 148, I do not understand how your results suggest an extinction debt in the north, as fig. 2 shows the high slopes of CTI in the north, and the sites there are assigned to the pathway of homogenization, which implies extinction of cold species.

Reply 2.5:

Again, we feel that the confusion was caused by some poor writing on our part. We have removed this mention of extinction debt in this section, but it is better explained later in the discussion (Lines 257-260). In brief, an extinction debt refers to the situation where multiple species are doomed to extinction without any further change in the environment. Here this is shown by the high extinction rates at the northern edges of the cold-adapted species distributions, suggesting that several species are on their way “out”. Naturally, this situation will worsen with even further changes in the environment, as our results show that many species are currently reaching the edge of their thermal optima, whereas temperatures continue to rapidly increase.

Lines 257-260: These deviations suggest that northern communities are currently experiencing the highest imbalance between extinctions and colonisations and that some of their species may currently be living outside of their preferred temperature range thus creating an extinction debt.

In l. 304-305 you talk about the assumption of neutrality of MacArthur and Wilson (1967). I doubt that they made such a claim. Their graphical model (in Figure 7 of the book) implies equivalence (in the sense that species identities are interchangeable for modeling purposes) but not independence (curved lines in Fig 7 imply a dependence of colonization and extinction rates with the number of species present, that is, species interactions). Although this can be considered neutrality in some sense (I do not fully agree, as their model implied species interactions even if we do not care of their identities),

it does not take into account the ultimate goal of MacArthur and Wilson of obtaining a “biogeography of the species” (as they call it in their prospect, chapter 8, and Wilson acknowledged in Losos and Ricklefs book). That being said, it is true that Alonso et al. 2015 and Ontiveros et al. 2019 assume species equivalence and independence, as a mathematical tool to facilitate parameter estimation (as Simberlof 1969 did to facilitate the mathematical treatment of the island biogeography model). So, I would say here just “under the assumption of species equivalence and independence, all species ...”.

Reply 2.6:

We have amended the sentence to match the Reviewer’s suggestion.

I encourage the authors to address these comments and improve their already interesting and sound manuscript.

Reply 2.7:

We are grateful to the Reviewer for their constructive and commendably specific suggestions, which have all been implemented in the revised text. This input has greatly improved the manuscript.

Reviewer 3

This contribution uses 30 years of macromoth collecting and reports a poleward shift of warm-adapted species and extinction of cold-adapted species in the north. For this process authors apply the concept of thermophilisation, prior mainly used in plant ecology to describe alpine gradients. Technically, the paper is well written and the methods are easy to understand. The topic fits into the scope of Nat. Comm.

Reply 3.1:

We would like to thank this Reviewer for their helpful suggestions, which have helped us improve the clarity of the manuscript and to evaluate the robustness of our results to multiple sources of uncertainty (as further detailed below).

Numerous prior studies have reported a poleward shift in various organisms, also in Lepidoptera. Consequently, southern populations are necessarily composed of relatively warm-loving species. Authors do not clearly explain why they use the thermal detour if they might simply report species poleward shifts. The respective reassembly of communities according to thermal adaptation is evident.

Reply 3.2:

While we appreciate this comment, we believe that it reflects a slight misunderstanding in what we were trying to achieve. Indeed, there are many studies (and reviews) showing poleward shifts among Finnish Lepidoptera (from <https://doi.org/10.1038/21181> through <https://doi.org/10.1111/j.1365-2486.2008.01789.x> to <https://doi.org/10.1093/evlett/grad004>), however, shifts in range limits was not a response considered in our contribution. Our paper adds a key insight into what has been previously lacking from this previous literature: a view into what is happening within actual, local communities. A shift in northern range limits

alone would namely suggest that local species richness simply increases – unless balanced by the extinction of other species. The question is then what species are added, what species disappear and how this reflects into the composition of the community. The potential scenarios are many, as outlined in our contribution (Figure 1). Assuming a general poleward shift across species will fail to resolve among them.

To resolve the mechanisms, we provide an exploration of thermal niche characteristics within communities. Addressing the changes in such characteristics is an emerging field of research, which has already provided novel insights into how climate change is driving community assembly (e.g. Montras-Janer et al., (2022): <https://doi.org/10.1038/s41559-024-02326-7>). Again, we stress that no part of our analysis is addressing actual shifts in distribution (such as shifts in northern distribution boundaries or points of gravity in ranges). Instead, we show how communities are warming under climate change, and that this is due to differential variation in colonisation-extinction dynamics of thermal affiliated groups of species along a latitudinal gradient.

Having said this, we understand that the two patterns (poleward shifts and shifts in local community composition) are closely connected. Yet, they are two sides of the same coin, observing one does not conclusively address the other. Indeed, poleward shifts could indirectly be inferred from our results, but characterising such patterns will tell us little about the community-level processes. We have now added extra detail to the methods, to make our current objectives clearer (Lines 274-277).

Lines 274-277: Lines 274-277: To understand whether communities are warming in response to climate change, we characterised the species in our observed communities by their thermal affinity - i.e. the species temperature index, STI (Schweiger et al., 2014), which is measured as the average temperature recorded in their European range).

A new aspect might be the selective extinction. Some species appear to be unable to adapt either to warmer temperatures or to altered resource availability. The study focuses on temperature only. However, higher temperatures alter resources (often host plants) and moth and host plant phenologies, sometimes leading to phenological mismatches. Both topics are not discussed. I think these points should be included at least in the discussion.

Reply 3.3:

We fully agree that several processes may contribute to local extinctions, but we aimed to keep our conclusions linked to the direct evidence from our data, which focuses only on the moth communities. The additional processes potentially contributing to the colonisation-extinction dynamics observed are discussed, as suggested by the Reviewer in Lines 228-234).

Lines 228-234: Overall, thermal affinity groups were characterised by different northern range limits, demonstrating a key impact of abiotic conditions in general and temperatures in particular on species membership in local communities. However, temperature is not the only factor limiting species occurrence. Individual species are oftentimes absent from substantial areas within their thermal niche range Moore et al. (2023), pointing to other factors limiting their distribution. Such factors involve dispersal abilities et al., 2024), phenological mismatches et al., 2021), species interactions (Louthan et al., 2015), habitat availability (Platts et al., 2019), and abiotic factors like precipitation or soil properties (Moore et al., 2023).

The present submission focuses on thermal niches, the range of temperatures a species occurs, and associated community wide temperature indices. Temperature range was calculated from the average temperatures across the occurrence range. This is a very raw approximation as mean temperature is not as decisive for Lepidoptera occurrence as, for instance, minimum temperatures and temperatures during activity periods. I wondered why the study did not use these more meaningful measures.

Reply 3.4:

While we fully agree that average temperatures will offer a crude proxy for biologically-relevant variation in thermal conditions, we stress two considerations: first, in the European context, mean temperature is linked to the annual minimum temperature and

temperatures across spring-summer (activity period) and hence it is a biologically meaningful and practical measure (see Schweiger et al., (2014) that direct assess this concern: 10.3897/zookeys.367.6185). Therefore, species' affinities with mean temperature can distinguish warm- and cold-affiliated species on a continental level and is the current consensus metric used for the metric that we call the Species Temperature Index (Schweiger et al., (2014): 10.3897/zookeys.367.6185). Of this, the Community Temperature Index (CTI) provides the community-level equivalent: it is simply the community-weighted mean STI. Importantly, mean temperature and its derivative STI has been identified as a strong predictor of community reassembly by a large body of research (Devictor et al., (2012): 10.1038/NCLIMATE1347, Montras-Janer et al., (2022): <https://doi.org/10.1038/s41559-024-02326-7>).

Following the Reviewer's point, we used another commonly-used temperature metrics to show that both mean temperature and growing degree days (GDD - a more biologically relevant metric) are highly correlated. We calculated GDD above a base temperature of 5°C, following the method by Kauppi and Posch (1988). We then compare our measure of mean temperature to the month-specific GDD values of February, April, June and August, we see that they are extremely well correlated ($r = 0.88-1.0$). As a result, our results are insensitive to the exact metric selected. In response, we have now included a justification for our choice of mean temperature (Lines 324-331) and added a Supplementary Figure S3: (copied below) showing the correlation matrix between this metric and other temperature measures.

Ref: (Kauppi P, Posch M (1988) A case study of the effects of CO₂-induced climatic warming on forest growth and the forest sector: A. Productivity reactions of northern boreal forests. In: Parry ML, Carter TR, Konijn N, editors. The Impact of Climatic Variations on Agriculture, Volume 1: Assessments in Cool Temperate and Cold Regions. Kluwer, Dordrecht, Netherlands. 183-195.)

Lines 324-331: To characterise the climatic niche of a species, we used the standard definition of STI i.e. the average mean temperature across its range (Schweiger et al., 2014). While a similar metric could be derived for other climatic descriptors, we found

mean temperature offered a good proxy for a range of metrics (as shown in Schweiger et al. (2014)). Across Europe, mean temperature shows high correlation ($r > 88\%$) with a range of other temperature measures (Supplementary Material Figure S3)

Figure S3: Correlation matrix of mean temperature (STI), and derived monthly growing degree day temperatures.

Further, temperatures increased during the last 30 years. What was the baseline for the STI calculation? Many species are known to have adapted to new temperature regimes for instance changing their activity periods. How was this included? I think the description and the rationale behind the STI assessment needs additional information.

Reply 3.5:

We would like to apologise for the lack of detail in our methods. In response, we now

provide more details about how STI was calculated. To account for the shift in baseline, we calculated STI by overlaying range maps with climatic means over 1961-1990, which is the time period just before our sampling started – and also the time period before accelerating climate change set in e.g. (Beaulieu et al.(2024) <https://doi.org/10.1038/s43247-024-01711-1>, Vicedo-Cabrera et al., 2021 <https://doi.org/10.1038/s41558-021-01058-x>). We agree that STI is unable to capture more nuanced processes shaping species’ ranges. However, STI can capture temperature affinities on a continental level and distinguish warm- and cold-affiliated species. As such, it offers a useful metric of species-specific distribution during comparable baseline conditions. Finally, the data collection process will account for any shifts in activity periods as the samples are collected during the full activity period of the majority of moth communities (March-September) and are aggregated by year. (Lines 319-322).

Lines 319-322: Based on species occupancy of grid cells across Europe, climatic niche metrics were calculated by overlaying range maps with climatic variables over the period between 1961 and 1990, which is the period just before our sampling started. Since this is also the period before accelerating climate change set in, it represents relevant baseline conditions

The description of the sampling needs clarification. In line 264 Macrolepidoptera were mentioned, while the main text speaks of Macromoths. Were butterflies included? The exclusion of singletons is unclear. Given the observed abundance gradients, particularly singletons might be of interest and an exclusion might bias the results.

Reply 3.6:

We apologise for the confusion and have double-checked and extended the description of our methods. Indeed, we only focused on macro moths (Lines 291-297).

Lines 291-297: In the current study, we focused on macro moths (Supplementary Material Table S7), since micro moths had been inconsistently scored from the samples. To ensure a robust and unbiased dataset for our analysis, we filtered the data for obvious sources of errors. Specifically, i) to confine our data to sites for which temporal change

can be reliably established, we chose sites with at least 10 years of data; and ii) to derive a comparable baseline between sites, we excluded any sites that started data collection later than 2005. These filtering processes resulted in 728 species in 62 sites.

The exclusion of singletons is unclear. Given the observed abundance gradients, particularly singletons might be of interest and an exclusion might bias the results.

Reply 3.7:

Importantly, most analyses of community warming (including parts of our analysis) are based on community-weighted mean STI. In such analyses, singletons carry little weight, as species are weighted by their abundance. In addition we have also clarified the exclusion of singletons (Lines 301-308, and Lines 215-226). The justification for excluding them is to lower the rate of pseudo-turnover (i.e., a species being present but going undetected; for an assessment of sample coverage, see below). To re-evaluate the impact of this choice, we have now re-run the analysis with singletons included vs. excluded from the data (see Supplementary Tables S4 and S5). We find that the resulting sets of colonisation and extinction rates (with and without singletons) are highly correlated ($r = 82-88\%$), and the results are qualitatively unchanged.

Lines 301-308: Since rare species are likely to be associated with low detection probabilities (and thus with high rates of pseudo-turnover (Alonso et al., 2015), we then explored whether our analysis was affected by the inclusion vs. exclusion of rare species. To this aim, we ran the analysis (see section Model fitting and inference below) with singletons (i.e. species observed as single individuals) included vs. removed from the dataset. We found that estimates of both colonisation rates and extinction rates were highly correlated between the two data sets ($r=0.82-0.88$, respectively). As there were no qualitative differences in results between the two sets, we report the results with singletons included in the main text, and results with singletons removed in Supplementary Material Tables S5 and S6.

In line 284 I learned that 728 species (out of a total of 1755 species) were used. Does

this mean that more than 50% of species were excluded? Or do the 1755 species refer to all Lepidoptera? In that case how many macromoths were collected and how many do occur in Finland? It might be that the cold adapted species did not die out but simply attained lower abundances due to resource availability.

Reply 3.8:

Again, we apologise for not being clear in our original description of the dataset. Based on this comment, we have now removed any mention of 1755 species from the text, as this number included sites not used, species of Micromoths (which were not consistently identified across sites and therefore removed from further analyses) etc. To clarify what exact set of species were included, we have extensively clarified the methods and data structure (See above **Reply 3.6** Lines 291-297)

I missed information about sample coverage. How many species are missing in the monitoring collections?

Reply 3.9:

To directly address this concern, we calculate sample coverage. This metric estimates what proportion of the community members (individuals) are represented by species detected in the sample (Chao et al. (2014): <https://doi.org/10.1890/13-0133.1>).

We calculated sample coverage using the iNEXT package Hsieh et al. 2016 (<https://doi.org/10.1111/2041-210X.12613>), following Chao et al. 2014 (<https://doi.org/10.1890/13-0133.1>). What we find is that our samples are consistently deep enough to attain over 99% species coverage. In fact, the communities seem to be frequently exhaustively sampled, and thus very well representative of the actual communities. Please see new additions to the text: Lines 297-300 and see Supplementary Material Table S7)

Lines 297-300: We then used iNEXT::iNEXT (Hsieh et al., 2016; Chao et al., 2014) to evaluate whether our sites exhibited sufficient coverage of the focal communities. Based

on sample completeness using rarefaction (Chao et al., 2014; Hsieh et al., 2016), all sites showed a sample-based coverage of >99% (Supplementary Material Table S4).

What about richness estimation for sites (rarefaction curves)? Did the different regions had comparable sample coverages?

Reply 3.10:

Following the Reviewer's insightful request for further assessment of sample coverage, we examined our data for regional variation in sample coverage. Indeed, sample coverage was consistently very high (and nearly complete), comparable among sites and regions, and showed no trend with latitude. See **Reply 3.9** above and **Reply 3.12** below. With a sample coverage above 99%, we infer that communities are representatively sampled and have therefore used species pools as they were observed.

I missed the raw data and respective summary tables for the single study regions (or traps?). Otherwise it is difficult to assess whether the results might stem from sample biases.

Reply 3.11:

All data are published along with the paper and accessible through Dryad: <http://datadryad.org/stash/share/g6NTS3eP2LGidOC0EFT4GovxGR4xmm4LIIJhUQA8yYA>

I guess that the overall lower abundances in northern regions also affect detection probabilities. Was there a latitudinal gradient in coverage and the frequency of singletons? In conclusion, the description of the data acquisition is insufficient.

Reply 3.12:

To provide a direct answer to this query, we re-examined our data, but found no evidence for a northwards decline in detectability: i) sample coverage was uniformly high

and showed no latitudinal pattern (see Reply 3.10), and ii) singletons were, in fact, more frequent in the south, providing further evidence against any northward decrease in detectability (see Figure below).

Figure Revision 1: The number of singleton species per site along a latitudinal gradient.

Finally, given the high species exclusion rates and possible detection issues I wondered whether extinction and colonisation times can be calculated reliably. These issues should be clarified.

Reply 3.13:

The impression of high exclusion rates was, unfortunately, caused by our poor data description (see Reply 3.8). Data selection included only standard pruning procedures, which have all been clarified in the revised text. We hope that our re-analysis of the impact of singletons and the extent of sample coverage (Replies 3.12, 3.9, 3.11 above), and the thorough overhaul of the description of data acquisition (see Replies 3.6 and 3.7) and other methods have sufficed to dispel any previous concerns.

Response to reviewer comments: *Extinction in the north and colonisation in the south: the latitudinal drivers of community warming.*

January 2025

Reviewer 1

Dear Authors, many thanks for the opportunity to review this work again. I look forward to reading more about this forthcoming trait-based analysis in the future. Happy to recommend for publication!

Reply 1.1:

Again, we would like to thank the Reviewer for their positive assessment and valuable input!

Reviewer 2

The authors have answered most of my concerns satisfactorily, but there are a few points where the reasoning appears to be inconsistent or even contradictory, which have emerged from their response.

Reply 2.1:

We are grateful to the Reviewer for their insightful and persistent help in making this the best possible contribution. We have carefully re-considered every aspect of the additional analysis introduced in our previous revision and have now introduced further changes to ensure that the reader gets a clear view of the colonisation-extinction patterns underpinning community warming.

In revisiting our inference, we realised that multiple factors should be highlighted to disentangle or infer thermophilisation pathways. Before, we were simply describing the equilibrium between extinctions and colonisations. Yet, to assess the net extinction rates (number of species lost per year), we should also (1) account for variation in the mean species richness of the thermal groups across bioclimatic zones and (2) adjust for species saturation (as reflecting potential colonists) – as it will define the magnitude of the colonisation rates *relative to* the number of species available for colonisation. We have now carefully reworked the relevant sections of the manuscript - (Figure 4), (Results lines: 127-152 & 163-173 and Discussion lines: 240-250 & 272-276).

1. The definition of the deviation between expected and observed richness in l. 131 contradicts the explanation in l.135-137 and Fig. 4A. If you define the deviation as $\log(\text{expected}/\text{observed})$, positive values imply a deficit of species (taking as a reference the expectation). To be perfectly clear, you can only get positive values if $\text{expected} > \text{observed}$, which I would call a deficit of species, and it is indicated as an excess of species in the text. Conversely, negative values would correspond to excess of species, while in the text it is indicated as a deficit. This needs to be clarified, because the interpretation of Fig 4a greatly varies if you interpret the results as a deficit or as an excess. Probably you just meant to define the deviation as $\log(\text{observed}/\text{expected})$.

Reply 2.2:

Not only do we agree with the reviewer, but also realise that our previous Figure 4A was over-simplified and difficult to interpret. This was evidenced by the authors' and the Reviewer's contrasting interpretations of the evidence presented.

In response, and as outlined above, we have now implemented a major overhaul of this section to provide a clearer assessment of the several components with a potential impact on extinction and colonisation dynamics across thermal affinity groups and space, thus providing transparent evidence for different pathways to thermophilisation. By exposing all the components of the equilibrium calculations (i.e. extinction and colonisation rates, species richness (mean), species saturation (potential colonists) and resulting net rates of colonisation and extinctions), we show the full picture and complexity of the patterns observed. This approach allows a full view of variation across thermal groups and latitudes, including the influence of species pools and the total abundance of the

communities. As suggested by the Reviewer, we have now re-written our discussion to clarify our inference and the nuances observed in the thermophilisation pathways (Discussion lines: 277-303).

2. If the interpretation is maintained in the same sense as it is portrayed in the text, I think it is not accurate to describe the thermophilisation pathway of the mid biozone as turnover. Turnover implies the colonization of warm species and extinction of cold ones, but Fig. 4A indicates that there is an excess of cold ones over expectation and a huge deficit of warm ones, which is opposite to the turnover pathway. I guess that it is still turnover, but not the one explained in Figure 1 (pathway 3). The interpretation of the other zones holds - in the south, you maintain (even have a few more) cold species and get some new warm ones, which is compatible with diversification, while in the north you have a deficit of cold species and maintain (no) warm ones, which is compatible with homogeneisation. Can you explain the mismatch in the mid biozone?

Reply 2.3:

We again appreciate the Reviewer's thorough evaluation of all the different aspects of our analysis and their insightful suggestions regarding turnover. In response, we now discuss how turnover is occurring throughout the different zones, as evidenced by both net colonisations and extinctions (Figure 4C, F) and how these turnover patterns can diverge from our idealised patterns in Figure 1 (Results lines 171-173 and Discussion lines 277-286).

Admittedly, there were only two data points (two sites) of warm-tolerant species in the mid-bioclimatic zone. This was obviously no problem in our previous analysis (where latitude was used as a continuous variable) – but for any discrete grouping of bioclimatic zones, we are clearly facing a scarcity of data. In response, we have opted to remove any warm-tolerant data from the mid-bioclimatic zone from the analysis. As stated above, and matching the Reviewer's concerns, we have also changed our narrative to suggest that turnover is occurring throughout the country, not just at mid-latitudes.

3. I am not really sure that the pathways, taken together and in the light of Fig 4, follow a logical succession. You could expect different pathways in different zones, sure, but getting diversification-turnover-homogeneisation in a latitudinal gradient does not seem intuitive. I would expect cold species to get extinct in the south and persist in the north, and warm species persist in the south and colonize the north. This would imply a homogeneisation-turnover-diversification sequence instead of the one shown currently in the article. Maybe you could comment on this and stress in the conclusions that the pathways you find are not the ones that may be expected a priori, if you deem it fit (it is not mandatory, though).

Reply 2.4:

We have now addressed this valid point by adding a new section to the discussion (lines 295-303). We attribute the discrepancies from our conceptual Figure 1 to the fact that our thermal affinity groups are based on the country-wide distributions of the STIs.

For example, the cold-adapted group corresponds to the lowest quartile of the STI distributions (Figure S4) and shows the largest range of STI values. Thus, their presence and persistence in the south is probably due to species at the upper STI-edge of the group being present in the south, whereas species with the coldest STIs of the cold-adapted group are being lost in the north. However, our current approach does not give us the full species-specific insight needed to test this. We have now directly identified this as a limitation of our paper. Had we held European-wide data on the extinction-colonisation dynamics of these groups, we should likely see them tracking their the climatic niche. For now, this is now suggested as a testable hypothesis for future studies – as necessitating a massively augmented data base.

I encourage the authors to solve these discrepancies (or at least acknowledge them) in order to have a rounder manuscript. I also thank them for their previous responses.

Reply 2.5:

We have now solved these discrepancies and hope that we have acknowledged any remaining concerns. With the help of the Reviewer, we feel that we have now arrived at clearer results of latitudinal and biotic drivers of community reassembly and the additional reflections of our limitations and nuances has a resulted in a rounder manuscript. Thanks you!

Reviewer 3

I was impressed by this revision. The authors fully answered to my concerns and questions and performed a number of reevaluations and additional analyses. They included the requested clarifications and extended particularly the methods section. I have no more comments.

Kind regards Werner Ulrich

Reply 3:

Dear Werner,

Thank you again for your helpful comments which have greatly enhanced the manuscript.